# Determination of Krogh Coefficient for Oxygen Consumption Measurement from Thin Slices of Rodent Cortical Tissue Using a Fick’s Law Model of Diffusion

**DOI:** 10.3390/ijms24076450

**Published:** 2023-03-29

**Authors:** D. Alistair Steyn-Ross, Moira L. Steyn-Ross, Jamie W. Sleigh, Logan J. Voss

**Affiliations:** 1School of Engineering, University of Waikato, Hamilton 3240, New Zealand; msr@waikato.ac.nz; 2Anaesthesia Department, Waikato District Health Board, Hamilton 3204, New Zealand; jamie.sleigh@waikatodhb.health.nz; 3Department of Anaesthesia, Waikato Clinical Campus, University of Auckland, Hamilton 3204, New Zealand; logan.voss@waikatodhb.health.nz

**Keywords:** Krogh coefficient, mouse cortical slice, diffusion–consumption equation, oxygen flux in brain tissue, thin-slice metabolism

## Abstract

To investigate the impact of experimental interventions on living biological tissue, ex vivo rodent brain slices are often used as a more controllable alternative to a live animal model. However, for meaningful results, the biological sample must be known to be healthy and viable. One of the gold-standard approaches to identifying tissue viability status is to measure the rate of tissue oxygen consumption under specific controlled conditions. Here, we work with thin (400 μm) slices of mouse cortical brain tissue which are sustained by a steady flow of oxygenated artificial cerebralspinal fluid (aCSF) at room temperature. To quantify tissue oxygen consumption (*Q*), we measure oxygen partial pressure (pO${}_{2}$) as a function of probe depth. The curvature of the obtained parabolic (or parabola-like) pO${}_{2}$ profiles can be used to extract *Q*, providing one knows the Krogh coefficient $K_{t}$, for the tissue. The oxygen trends are well described by a Fick’s law diffusion–consumption model developed by Ivanova and Simeonov, and expressed in terms of ratio (Q/K), being the rate of oxygen consumption in tissue divided by the Krogh coefficient (oxygen diffusivity × oxygen solubility) for tissue. If the fluid immediately adjacent to the tissue can be assumed to be stationary (i.e., nonflowing), one may invoke conservation of oxygen flux K·(∂P/∂x) across the interface to deduce $\left(K_{t}/K_{f}\right)$, the ratio of Krogh coefficients for tissue and fluid. Using published interpolation formulas for the effect of salt content and temperature on oxygen diffusivity and solubility for pure water, we estimate $K_{f}$, the Krogh coefficient for aCSF, and hence deduce the $K_{t}$ coefficient for tissue. We distinguish experimental uncertainty from natural biological variability by using pairs of repeated profiles at the same tissue location. We report a dimensionless Krogh ratio $(K_{t}/K_{f})\;=\;0.562\pm 0.088$ (mean ± SD), corresponding to a Krogh coefficient $K_{t}=\left(1.29\pm 0.21\right)\times 10^{-14}$ mol/(m·s·Pa) for mouse cortical tissue at room temperature, but acknowledge the experimental limitation of being unable to verify that the fluid boundary layer is truly stationary. We compare our results with those reported in the literature, and comment on the challenges and ambiguities caused by the extensive use of ‘biologically convenient’ non-SI units for tissue Krogh coefficient.

## 1. Introduction

The acute ex vivo brain slice technique [1] is widely used by electrophysiologists as an experimental tool for probing neurophysiological health and disease. Because brain slices lack a blood flow, a continuous supply of glucose and oxygen must enter the tissue by diffusion from the perfusing liquid that serves as an artificial cerebral spinal fluid (aCSF) to maintain cell vitality. This slow diffusion of essential elements, combined with inevitable mechanical damage by the sectioning procedure, compromises the viability of the bulk tissue. As a result, for slice experimentalists in general (and particularly those wishing to follow the time course of tissue repair and regeneration), it is essential to have an objective and validated method for quantifying the health of test slices. Electrophysiological methods are commonly used to assess slice viability [2,3]. However, functional electrophysiological output, while helpful, does not provide a direct or complete readout of tissue status [4]. A cleaner approach would be to measure tissue oxygen consumption—healthier tissue consumes more oxygen [5,6,7]—providing a direct signal of cell viability.

Oxygen consumption can be quantified from thin sections of living tissue by profiling oxygen tension (partial pressure) as a function of tissue depth, probing from the upper surface to the lower surface. The curvature of the pressure vs. depth profile allows extraction of *Q*, the rate of oxygen consumption per unit volume of tissue, provided one knows the Krogh coefficient, $K_{t}$, for the tissue. The Krogh coefficient is a lumped constant that describes oxygen permeability, being the product of oxygen diffusion coefficient (*D*) and oxygen solubility (*S*). Because oxygen solubility is difficult to measure in metabolising tissue [8], the $K_{t}$ lumped constant is usually reported for active tissue experiments. A wide range of values of Krogh coefficient for biological tissue appear in the literature [8,9,10,11,12,13,14,15], so it is not immediately clear which value to choose. In addition, most reported values assume a bath temperature of 37 ${}^{{}^{\circ}}C$, while many experiments (including those reported here) are run at room temperature (20–25 ${}^{{}^{\circ}}C$). The fluid Krogh coefficients $K_{f}$ for pure water and saline are known to increase with temperature, so it is very probable that a temperature correction for $K_{t}$ would also be required. For these reasons, we sought to derive a tissue $K_{t}$ value specific to our experimental conditions and, in so doing, provide a unified theoretical and experimental framework for deriving $K_{t}$ for other experimental models.

A complicating factor when attempting to compare metabolic studies is the fact that at least six different compound units for Krogh coefficient are in common use (see Table 1 for a survey), and while these are all metric combinations, none adhere to the SI standard. This unfortunate state of affairs makes study comparisons challenging since multiple unit interconversions are required, leading to potential ambiguity and error. In this paper, our measurements for oxygen partial pressure [mmHg] and probe depth [μm] are dictated by the instrumentation at hand (Clark-style oxygen electrode and micromanipulator, respectively), but we ensure that our final Krogh results are either quoted in ratio form $\left(K_{t}/K_{f}\right)$ [tissue:fluid Krogh ratio, dimensionless] or using standard SI [mol/(m·s·Pa)].

The goals in this paper are twofold. Building on earlier work by Ivanova & Simeonov (2012) [10], our first goal is to provide a unified mathematical foundation for the experimental determination of the Krogh coefficient in a thin slice of metabolically active tissue sustained by a continuous flow of oxygenated fluid. Our second is to demonstrate application of the theory to slices of mouse cortex in order to quantify error bounds on the Krogh coefficient, and to identify limitations in the theory. Our overarching motivation is to provide a strong theoretical and experimental basis for quantifying oxygen consumption in mouse cortical slices, thereby allowing clear and unambiguous classification of tissue viability status.

The paper is structured as follows. In Section 2.1, we present the classical 1D Fick’s law partial differential equation (PDE) model to describe the concentration-driven diffusion of oxygen through a thin slab of metabolically active tissue. At steady-state, the oxygen concentration and gas pressure (tension) at a given depth *x* will be unchanging; so, the PDE reduces to a second-order differential equation in oxygen tension *P* whose curvature is proportional to *Q*, the rate of oxygen consumption per unit volume of tissue. By selecting an idealised piecewise-linear model for consumption rate (Section 2.2), the differential equation can be solved exactly. In Section 2.3, we list the Ivanova & Simeonov [10] solutions for three distinct boundary conditions, but choose here to work with dimensioned quantities, and to show explicitly the mirror symmetry of the *P* vs. *x* solutions about the slice central axis at x=0. In Section 2.4, we demonstrate the validity of these solutions via optimised curve fits to four representative oxygen profiles obtained from cortical slices sustained in our perfusion bath (described later in Section 3.1).

Section 2.5 surveys the surprisingly wide range of non-SI but ‘biologically convenient’ units used for Krogh coefficient in the literature, and comments on the potential ambiguity that can arise for unit conversions involving $V_{m}$, the molar volume of oxygen. In Section 2.6, we use literature sources to investigate the temperature dependence of the Krogh coefficient of pure water, then consider the effect on $K_{f}$ of adding salts and glucose to create aCSF, the artificial cerebrospinal fluid. We conclude the theoretical discussion in Section 2.7 by describing how the tissue Krogh coefficient $K_{t}$ can be determined via flux conservation across the tissue–fluid boundary.

Materials and Methods (Section 3) details how the slices of brain tissue are prepared and sustained with a steady flow of oxygenated aCSF in the perfusion bath (Section 3.1). Section 3.2 describes the experimental setup that allows dual-hemisphere recording of both LFP (local-field potential) electrical activity and oxygen-tension variation with depth using a pair of oxygen probes co-located with the LFP wire electrodes. Our standard protocol (Section 3.3) is to measure oxygen tension as a vertical profile, with soundings taken every 50 μm through the fluid and tissue. The resulting pO${}_{2}$ pressure profiles are processed using custom-written Matlab software (Section 3.4) to locate the slice centre, then extract the ($Q/K_{t}$) curvature via iterative curve fitting. If a stationary fluid layer is detected, then the Krogh ratio $\left(K_{t}/K_{f}\right)$ can be determined, and hence an estimate for $K_{t}$ for a given pO${}_{2}$ profile.

We present our results in Section 4 for a range of fluid flow rates. The statistics for our $K_{t}$ determinations show a large scatter about the mean value. By using pairs of repeated profiles at the same tissue location, we are able to distinguish natural biological variability from true experimental uncertainty. Our results suggest that, when comparing profiles from different locations within the same slice, natural variability in tissue Krogh coefficient is about three times larger than experimental errors arising from measurement uncertainty, curve fitting, and boundary gradient calculations. In Section 5, we discuss our findings, and compare our results with Krogh coefficient values reported by other workers. We acknowledge limitations in our experimental approach, and make suggestions for future work.

## 2. Theoretical Background: Model Equations and Solutions

### 2.1. Diffusion Equations

Krogh’s foundation paper of 1919 [16] examined oxygen flow from a blood capillary source to surrounding metabolically active muscle tissue. He assumed an ideal two-dimensional (2D) geometry in which a cylindrical capillary of radius *r* is enclosed within a concentric outer cylinder of tissue of radius R>r, then solved the 2D Fick’s law of diffusion to give an expression for oxygen tension as a function of distance from the centre of the capillary.

For our ex vivo slice preparation, there is no blood supply. Instead, a steady flow of oxygenated perfusion fluid passing above and below the slice provides the tissue with oxygen via diffusion from the fluid to the tissue. The slice of brain material can be treated as a horizontal plane sheet of homogeneous tissue sufficiently thin that all of the diffusing oxygen enters through the top and bottom plane faces with a negligible amount through the edges (see Figure 1 for geometry). Oxygen concentration is then a function of vertical displacement from the centre of the slice, and the problem becomes one-dimensional.

We wish to derive an expression describing the steady-state condition in which the local diffusive supply of oxygen exactly balances the local metabolic demand. Our derivation of Equation (Equation 4) follows Ganfield et al. (1970) [8] but uses updated naming conventions for symbols.

The concentration *C* of oxygen [SI units: mol/m${}^{3}$] in the tissue is modelled as a 1D Fick’s law diffusion–consumption process,
(1)$$\begin{array}{c} \frac{\partial C}{\partial t}\;=\;D\frac{\partial^{2}C}{\partial x^{2}}-Q\left(C\right) \end{array}$$
where *D* is the diffusion coefficient [m${}^{2}$/s] of oxygen in tissue, and *Q* is the rate of oxygen consumption [mol/(m${}^{3}\cdot$ s)]. From Henry’s gas law, oxygen concentration *C* in the tissue is proportional to local oxygen partial pressure (tension) *P* [Pa] with proportionality constant *S* [mol/(m${}^{3}\cdot$ Pa)] being the gas solubility (also known as Henry’s law constant *H*) in tissue, with
(2)$$\begin{array}{c} C\;=\;P\;S \end{array}$$
allowing (Equation 1) to be rewritten in terms of oxygen tension *P*,
(3)$$\begin{array}{c} S\;\frac{\partial P}{\partial t}\;=\;DS\;\frac{\partial^{2}P}{\partial x^{2}}-Q\left(P\right) \end{array}$$

At steady state, the oxygen concentration (and pressure) is unchanging. Setting the left-hand side of (Equation 3) to zero gives the steady-state condition that diffusive oxygen supply in the tissue matches local metabolic demand,
$$\begin{array}{c} DS\;\frac{\partial^{2}P}{\partial x^{2}}\;=\;Q\left(P\right) \end{array}$$
which we rewrite as,
(4)$$\begin{array}{c} K\;\frac{\partial^{2}P}{\partial x^{2}}\;=\;Q\left(P\right) \end{array}$$
where K≡DS is the Krogh diffusion coefficient [mol/(m·s·Pa)], a lumped constant giving the permeability of oxygen in tissue.

Following Ivanova & Simeonov [10], we define oxygen flux *q* (rate of oxygen flow per unit area of tissue) [mol/(m${}^{2}\cdot$ s)],
(5)$$\begin{array}{c} q\left(x\right)\;=\;K\;\frac{\partial P}{\partial x} \end{array}$$
(Strictly speaking, the right-hand side of (Equation 5) should carry a minus sign indicating that oxygen flows in the direction in which pressure *decreases*, but for simplicity we follow the Ivanova & Simeonov convention, and treat flux as an unsigned quantity).

Assume the tissue slice has thickness 2L. The goal is to write down analytic expressions for oxygen consumption Q(x) and flux q(x) as a function of tissue depth *x* (with −L≤x≤L), where x=0 locates the centre of the slice.

Ivanova & Simeonov [10] have solved differential Equations (Equation 4) and (Equation 5) by assuming an idealised form (Figure 2) for the Q(P) consumption vs. tension profile, then applying appropriate boundary conditions.

**Figure 2 ijms-24-06450-f002:**
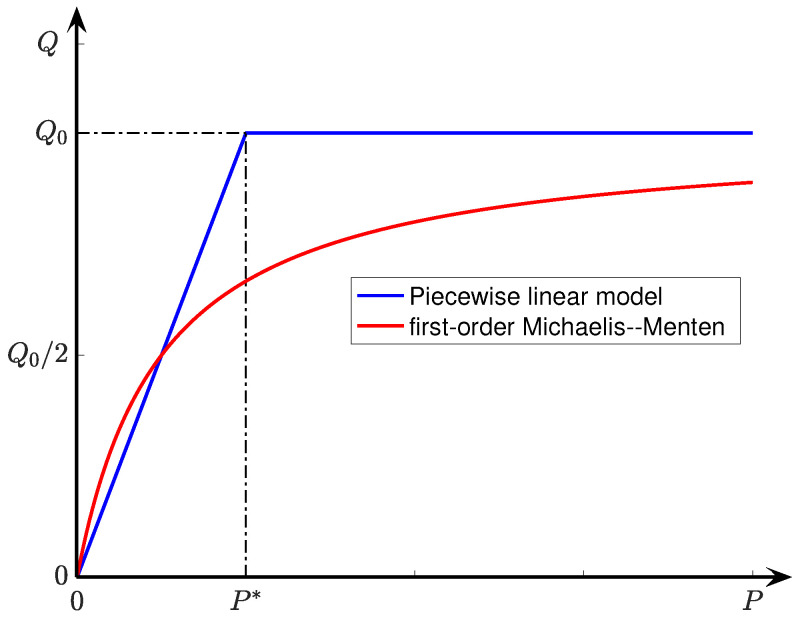
Two models for oxygen consumption rate in tissue as a function of oxygen tension. Ivanova and Simeonov’s piecewise-linear model of Equation (Equation 6) (blue curve) approximates the first-order Michaelis–Menton form $Q\left(P\right)=Q_{0}\;P/\left(K_{M}+P\right)$ with Michaelis constant $K_{M}=P^{*}/2$ (red curve).

Their solutions take the form of pressure–depth (*P* vs. *x*) profiles that are symmetric about x=0, and take one of three forms:(a)simple parabolic if minimum tension $P_{\min}$ exceeds a critical tension value $P^{*}$;(b)mixed parabolic/hyperbolic-cosine (cosh) if the P(x) profile crosses the $P^{*}$ boundary;(c)hyperbolic-cosine if the tension at the x=L tissue surface, $P_{s}$, is less than $P^{*}$.

Once the theoretical pressure profile has been established, the oxygen consumption rate can be determined—provided the Krogh coefficient *K* (tissue oxygen permeability) is known or can be assumed.

Because their solutions provide the foundation for the metabolic analyses reported here, we summarise the Ivanova and Simeonov (I & S) model and their mathematical solutions in the following two subsections.

### 2.2. Model for Oxygen Consumption vs. Oxygen Partial Pressure

Ivanova and Simeonov [10] treat respiring tissue as a continuous medium with distributed oxygen consumption that exhibits two behaviours: (i) If the local oxygen tension exceeds a critical pressure $P^{*}$, then consumption is maximal with $Q=Q_{0}$, a constant. In this regime, oxygen supply matches metabolic demand. (ii) If, however, oxygen tension falls below the critical value, $P<P^{*}$, then consumption drops off linearly with pressure such that Q=0 at P=0. This gives the I & S piecewise-linear consumption–pressure model,
(6)$$\begin{array}{c} Q\left(P\right)\;=\;\left\{\begin{array}{cc} \left(P/P^{*}\right)\;Q_{0}, & \;0\leq P\leq P^{*}\\ \;Q_{0}, & \;P>P^{*} \end{array}\right. \end{array}$$
illustrated in Figure 2. This simple piecewise form provides an elegant approximation to more complicated curvilinear relationships such as Michaelis–Menten and its generalisations,
(7)$$\begin{array}{c} Q\left(P\right)\;=\;Q_{0}\;\frac{P^{n}}{K_{M}^{n}+P^{n}} \end{array}$$
illustrated for the n=1 case in Figure 2. The Michaelis constant $K_{M}$ is the oxygen pressure corresponding to half-maximum oxygen consumption: $Q\left(K_{M}\right)=Q_{0}/2$.

The significant advantage of the Ivanova and Simeonov piecewise form (Equation 6) is that it permits exact analytic solutions of the diffusion and flux differential Equations (Equation 4) and (Equation 5). In addition, pleasingly, we find that the I & S solutions provide a good match with the pressure profiles we measured within thin (400 μm) slices of mouse cortical tissue.

### 2.3. Solution of Ivanova & Simeonov Diffusion Equations

Oxygen flux is directed from the oxygenated aCSF (artificial cerebral spinal fluid) into the tissue depth through the surface boundary (x=±L) marking the upper and lower interfaces between tissue and the surrounding perfusion fluid. Oxygen diffusion and consumption within the tissue are controlled by the oxygen tension $P_{s}$ at the tissue surface. Because of the compound nature of the I & S oxygen consumption model (Equation 6), there are three distinct cases to consider:(a)if the surface pressure $P_{s}$ is sufficiently high such that tissue pressure everywhere exceeds the critical value, i.e., $P\left(x\right)>P^{*}$, then oxygen consumption rate is constant throughout the tissue: $Q\left(x\right)=Q_{0}$;(b)if the surface oxygen tension is insufficiently high, then the declining internal tissue pressure may reach the critical value at some depth x=±δ, i.e., $P\left(|\delta |\right)=P^{*}$, in which case we separate the tissue into ‘outer’ layers (δ<x≤L) and (−L≤x<−δ) which are well supplied with oxygen so that $Q\left(x\right)=Q_{0}$, and an ‘inner’ layer (−δ≤x≤δ) with linearly restricted oxygen supply: $Q\left(P\right)=\left(P/P^{*}\right)Q_{0}$;(c)if the surface pressure lies *below* the critical value, $P_{s}<P^{*}$, then $P\left(x\right)<P^{*}$ everywhere in the slice, and consequently consumption rate scales linearly with pressure: $Q\left(P\right)=\left(P/P^{*}\right)Q_{0}$.

These three cases result in three different tension–depth profiles as illustrated in Figure 1. Note that, because of the assumed slab symmetry of the 1D slice, the pressure curves are all symmetric about the slice centre at x=0. The three sets of I & S solutions are itemised below:


**Case (a):**

$P\left(x\right)>P^{*}\Rightarrow Q\left(x\right)=Q_{0}=\mathrm{const}$


(8a)
$$\begin{array}{cc} P(x)\; & =\;P_{s}-\frac{Q_{0}}{2K}\left(L^{2}-x^{2}\right),\;-L\leq x\leq L \end{array}$$


(8b)
$$\begin{array}{cc} q(x)\; & =\;Q_{0}x \end{array}$$

 This is the simplest case. The tension profile is parabolic throughout, and the flux profile is linear with zero flux at the slice centre.

**Case (b):** Profile crosses critical value at depth x=±δ: i.e., $P\left(|\delta |\right)=P^{*}$For inner layer (−δ≤x≤δ):
(9a)$$\begin{array}{cc} P(x)\; & =\;P^{*}\;\frac{\cosh(\alpha x)}{\cosh(\alpha \;\delta )},\;\mathrm{with}\;\alpha \equiv \sqrt{\frac{Q_{0}}{P^{*}K}} \end{array}$$
(9b)$$\begin{array}{cc} q(x)\; & =\;K\cdot \left(\alpha P^{*}\;\frac{\sinh(\alpha x)}{\cosh(\alpha \;\delta )}\right) \end{array}$$For outer layer (δ<x≤L):
(10a)$$\begin{array}{cc} P(x)\; & =\;P^{*}\;+\;\left[\alpha P^{*}\tanh\left(\alpha \;\delta \right)\right]\left(x-\delta \right)\;+\;\frac{Q_{0}}{2K}{\left(x-\delta \right)}^{2} \end{array}$$
(10b)$$\begin{array}{cc} q(x)\; & =\;K\cdot \left(\;\alpha P^{*}\tanh\left(\alpha \;\delta \right)\;+\;\frac{Q_{0}}{K}\left(x-\delta \right)\;\right) \end{array}$$ These equations for pressure and flux also apply to the mirror-image outer layer (−L≤x<−δ) after making the substitution (δ→−δ) in Equation (10). Note that the value of δ in Equations (9) and (10) is unknown in advance, and must be determined numerically by solving the nonlinear equation,
(11)$$\begin{array}{c} \tanh\left(\alpha \;\delta \right)\;=\;\frac{1-P_{s}/P^{*}}{\alpha (\delta -L)}\;+\;\frac{1}{2}\alpha \left(\delta -L\right) \end{array}$$

**Case (c):**$P_{s}<P^{*}\Rightarrow$ entire profile lies below critical oxygen tension
(12a)$$\begin{array}{cc} P(x)\; & =\;P_{s}\;\;\frac{\cosh(\alpha x)}{\cosh(\alpha L)},\;-L\leq x\leq L \end{array}$$
(12b)$$\begin{array}{cc} q(x)\; & =\;K\cdot \left(\alpha P_{s}\;\frac{\sinh(\alpha x)}{\cosh(\alpha L)}\right) \end{array}$$

Note that Ivanova & Simeonov present their theoretical solutions in terms of *dimensionless* lengths ($x_{n}\equiv \alpha x$, $L_{n}\equiv \alpha L$, $\delta_{n}\equiv \alpha \delta$), whereas we have chosen to work with dimensioned length quantities (*x*, *L*, δ). With this change in notation, our Equations (8)–(12) are fully equivalent to their solutions (15)–(25).

A significant feature of these equations is that consumption rate $Q_{0}$ and Krogh coefficient *K* always appear as a lumped ratio, $\left(Q_{0}/K\right)$; this is an inevitable consequence of the way diffusion Equation (Equation 4) has been set up. This means that, using oxygen pressure measurements taken entirely within the slice, it is not possible to determine $Q_{0}$ unless the *K* is known, and vice versa. However, if there is a stationary boundary layer of fluid immediately adjacent to the tissue surface, then we can assume conservation of oxygen flux across the surface (from fluid to tissue at x=L), and use the known value of Krogh coefficient of water to determine the Krogh coefficient for tissue. This idea is developed further in Section 2.7.

The ratio $\left(Q_{0}/K\right)$ [SI: Pa/m${}^{2}$] has a physical meaning. For the pure parabolic profile (8) of Figure 1a, $\left(Q_{0}/K\right)$ is the *curvature* of the pressure parabola at the x=0 midpoint of the tissue slab. Suppose the pressure curve is fitted to the polynomial $P\left(x\right)=ax^{2}+c$, then the quadratic coefficient, doubled, gives the curvature: $(Q_{0}/K)=2a$. More generally, $\left(Q_{0}/K\right)$ can be thought of as the *metabolic rate per unit permeability* for oxygen in tissue.

### 2.4. Representative Oxygen Tension Profiles

Figure 3 shows four representative sets of oxygen partial-pressure measurements obtained from 400-μm slices of mouse cortical tissue using the methods described in Section 3; note that, compared with the exemplar traces of Figure 1, the graph axes here have been rotated 90${}^{{}^{\circ}}$ counterclockwise.

Each dataset has been fitted to either the parabolic profile of Equation (8) as shown in Figure 3a, or to the mixed parabolic-cosh-parabolic profile of Equations (9) and (10) (Figure 3b). Oxygen tension reaches its minimum value at the centre of the slice. Typically, when the flow rate of the perfusing aCSF is reduced, oxygen tension at the tissue/fluid boundary is reduced, and the entire profile is displaced towards the zero-pressure axis. The flattened curves in panel (b) are an indicator of insufficient oxygen tension to satisfy local metabolic demand.

### 2.5. Choice of Units: SI vs. ‘Biological’

As described in Section 2.1, the Krogh coefficient quantifies the permeability of oxygen in tissue, with SI units mol/(m·s·Pa) representing (molar amount of gas) per (unit distance in tissue × unit time × unit pressure). In the physiology literature, we find that it is common for authors to use a wide range of ‘biologically convenient’ (non-SI) units for the Krogh coefficient. Table 1 illustrates the dimensional diversity for choice of units with length (cm, mm); time (min, s); pressure (atm, mmHg, torr); gas amount (mmol, nM·mm${}^{3}$, cm${}^{3}$, mL).

While it is straightforward to convert between units of length, time, and pressure, e.g.,
1atm=101.325kPa=760mmHg=760torr
the conversions between the various units for gas amount (mmol, cm${}^{3}$, mL) require that we know the value of the gas molar volume at a specified temperature and pressure. If the author has not stated the temperature and pressure conditions for the metabolic measurements, then the gas molar volume is unknown. One might consider assuming IUPAC *standard* temperature and pressure (STP) conditions, or, alternatively, the NIST definition for *normal* temperature and pressure (NTP), and then apply ideal gas theory to compute $V_{m}$, the oxygen volume per mole,
$$pV\;=\;nRT\;\Rightarrow \;V_{m}\;=\;\frac{RT}{p}$$
where R=8.314462 J/(mol·K) is the universal gas constant. The resulting STP and NTP values for $V_{m}$ for an ideal gas are listed in Table 2; note that the IUPAC definition for STP changed in 1982. We observe that the molar volume at NTP is ∼10% larger than that for either definition of STP.

Of course, real gases are only approximately ideal. The measured molar volume for oxygen (at T=273.15K, p=1 atm) is 22.392 L/mol ([17] [p. 329, Table 6]), about 0.1% below the ideal gas value of 22.414 L/mol.

### 2.6. Krogh Coefficient for Oxygen in Water

In our slice experiments (see Section 3 for details), oxygen is delivered to the tissue-slice via diffusion from the surrounding no-Magnesium (no-Mg) aCSF perfusate. Table 3 lists its chemical composition. The no-Mg fluid contains glucose; three chloride salts (NaCl, KCl, CaCl${}_{2}$); and pH buffering agents (NaHCO${}_{3}$, HEPES). We assume the buffering agents have negligible impact on oxygen transport and solubility.

**Table 3 ijms-24-06450-t003:** Chemical composition of Normal and no-Magnesium (no-Mg) artificial cerebrospinal fluids (aCSF) used in mouse cortical thin-slice experiments. Last column lists chloride concentrations for the no-Mg fluid. Total chloride concentration Σ=4.928 g/L corresponds to chlorinity $\left[m_{\mathrm{Cl}}\right]$ = 4.904 g/kg.

Species	Molar Mass	Normal aCSF	no-Magnesium aCSF		Chloride Content
	(g/mol)	(mM)	(mM)	(g/L)	(g/L)
NaCl	58.44	130	130	7.597	4.609
KCl	74.55	2.5	5	0.373	0.177
MgCl${}_{2}$	95.21	1	—	—	—
CaCl${}_{2}$	110.98	2	2	0.222	0.142
NaHCO${}_{3}$	84.01	2.5	2.5	0.210	
NaOH	40.00	3.5	3.5	0.140	
HEPES	238.30	10	10	2.383	
D-glucose	180.16	20	20	3.603	
					Σ = 4.928

Relative to pure water, the glucose content of aCSF is expected to increase fluid viscosity slightly, thereby reducing gas diffusivity within the fluid. The presence of dissolved salts will reduce oxygen solubility (‘salting-out’ effect) by an amount proportional to total chlorinity (chloride content expressed in g of Cl${}^{-}$ per kg of solution). In addition to these direct solute effects, both O${}_{2}$ transport and solubility vary with temperature but in opposite directions: O${}_{2}$ diffusivity increases moderately strongly with temperature, while O${}_{2}$ solubility decreases less strongly, so the Krogh coefficient, being the product of diffusion and solubility, is expected to increase weakly as fluid temperature increases.

#### 2.6.1. Diffusion Coefficient for Oxygen in Water

Han and Bartels (1996) [18] give an interpolation formula for the temperature dependence of oxygen diffusion [in cm${}^{2}$/s] in pure water. We convert to SI units [m${}^{2}$/s] by adjusting their base-10 exponential offset from −4.410→−8.410, and their formula now reads,
(13)$$\begin{array}{c} \log_{10}\left[\frac{D(T)}{m/s{}^{2}}\right]\;=\;-8.410+\frac{773.8}{T}-{\left(\frac{506.4}{T}\right)}^{2} \end{array}$$
for temperature *T* in kelvin. The resulting O${}_{2}$ diffusion curve D(T) is plotted in Figure 4a.

#### 2.6.2. Solubility of Oxygen in Water

For oxygen solubility in pure water and saline solutions, Green and Carritt (1967) [19] provide an interpolation expression that shows explicit dependence on both fluid temperature and chloride content,
(14)$$\begin{array}{cc} \alpha \; & =\;10^{-3}\exp\{\left(-7.424+\frac{4417}{T}-2.927\;\ln T+0.04238\;T\right)\;-\\ \; & \;\left[m_{\mathrm{Cl}}\right]\left(-0.1288+\frac{53.44}{T}-0.04442\;\ln T+7.145\times 10^{-4}\;T\right)\} \end{array}$$
where α is the Bunsen absorption coefficient of oxygen, i.e., the volume of gas ($T=0{\;}^{{}^{\circ}}C$ and p=1 atm) dissolved per unit volume of solvent when the partial pressure of oxygen is 1 atm; $\left[m_{\mathrm{Cl}}\right]$ is the chlorinity, defined as the amount of chloride in grams per 1000 g of solution. Expression (Equation 14) has been used by Stroe and Janssen (1993) [20] (but note their Equation (5) has omitted ‘*T*’ in the final term in the chlorinity component), and the chlorinity expression of (Equation 14) is recommended by Forstner and Gnaiger (1983) [17].

To convert (Equation 14) from (volume ratio per atm) to oxygen solubility *S* in SI units [mol/(m${}^{3}$·Pa)], we write
(15)$$\begin{array}{cc} \frac{S\left(T,[m_{\mathrm{Cl}}]\right)}{\mathrm{mol}/(m{}^{3}\cdot \mathrm{Pa})}\; & =\;\alpha \times \left(\frac{1\;\mathrm{atm}}{101\;325\;\mathrm{Pa}}\right)\left(\frac{1\;\mathrm{mol}}{22.392\;L}\right)\left(\frac{10^{3}\;L}{1\;m^{3}}\right)\;=\;4.4075\times 10^{-4}\;\alpha \end{array}$$
where the (22.392L) divisor is the (non-ideal) molar volume for O${}_{2}$ at 0 ${}^{{}^{\circ}}C$, 1 atm. The oxygen solubility curve for pure water and for no-Mg aCSF saline (chlorinity $[m_{\mathrm{Cl}}]=4.904$ g/kg) are illustrated in Figure 4b.

A recent NASA–JPL report ([21] [Table 5.5, pp. 5–162] on atmospheric data tabulates a list of Henry’s law constants [mol/(L·atm)] for pure water. After converting to SI, the entry for oxygen reads,
(16)$$\begin{array}{c} \frac{S(T)}{\mathrm{mol}/(m{}^{3}\cdot \mathrm{Pa})}\;=\;9.8692\times 10^{-3}\exp\left[-161.6+\frac{8160}{T}+22.39\;\ln T\right] \end{array}$$

We compared S(T) of (Equation 16) against the zero-chlorinity limit S(T,0) of (Equation 15) for the temperature range $0\leq T{/}^{{}^{\circ}}C\leq 50$, finding excellent agreement with an rms difference of ∼0.72%. We choose to use (Equation 15) for our aCSF solubility modelling because of its convenient inclusion of salting-out effects.

For completeness, we also list the two-parameter expression for O${}_{2}$ solubility in pure water extracted from the comprehensive online database of Henry’s law constants maintained by Sander (2015) [22],
(17)$$\begin{array}{c} \frac{S(T)}{\mathrm{mol}/(m{}^{3}\cdot \mathrm{Pa})}\;=\;S_{\mathrm{ref}}\;\exp\left[1500\left(\frac{1}{T}-\frac{1}{T_{\mathrm{ref}}}\right)\right] \end{array}$$
where $S_{\mathrm{ref}}=1.3\times 10^{-5}$ mol/(m${}^{3}$·Pa) at reference temperature $T_{\mathrm{ref}}=298.15$ K (25 ${}^{{}^{\circ}}C$). As expected, this two-parameter, two-sig-fig fit is less accurate than Green and Carritt (Equation 15) (rms difference ∼2.6% for [0–50] ${}^{{}^{\circ}}C$ ), but has the advantage of direct interpretation of its numerical coefficients.

#### 2.6.3. Glucose Effects on Oxygen Diffusion and Solubility

Stroe-Biezen et al. (1993) (see Figures 7, 8 of [23]) quantified the effect of dissolved glucose on oxygen transport in water at 25 and 37 ${}^{{}^{\circ}}C$ for glucose concentration $C_{g}$ ranging from 0 to 1 mol/L. Their O${}_{2}$ diffusion and O${}_{2}$ concentration graphs reveal slow monotonic decays as glucose content is increased. By interpolating for $C_{g}=0.02$ mol/L (see Table 3), we are able to estimate the glucose depression effects for our no-Mg perfusion fluid:O${}_{2}$ diffusion coefficient: 1% depressionO${}_{2}$ solubility: 0.5% depression

We assume these effects are temperature independent, but also cumulative when computing the Krogh permeability for oxygen in aCSF, leading to a net 1.5% depression below the saline curve as illustrated in Figure 4c.

Our brain-slice experiments were run at room temperature which typically lay within 20 to 25 ${}^{{}^{\circ}}C$. Reading off the aCSF curve (dashed-red trace) of Figure 4c, this temperature range corresponds to Krogh permeabilities [2.262 to 2.348] $\times 10^{-14}$ mol/[m·s·Pa], suggesting a reasonable working value for $K_{f}$, the Krogh coefficient for no-Mg aCSF at room temperature is
(18)$$\begin{array}{c} K_{f}\;=\;\left(2.30\pm 0.04\right)\times 10^{-14}\;\mathrm{mol}/\left[m\cdot s\cdot \mathrm{Pa}\right] \end{array}$$

### 2.7. Krogh Coefficient for Tissue via Flux Conservation

In their oxygen profile measurements for a cat cerebral cortex, Ganfield et al. (see Figure 4 in [8]) observed a linear decrease in O${}_{2}$ tension across a narrow layer (∼40 μm) of solution immediately above the surface of the tissue, indicating a thin boundary layer of nonflowing solution exists above the tissue slice. Equating the oxygen flux $q_{f}$ crossing this fluid layer to $q_{t}$, the flux crossing the tissue surface, and the tissue Krogh coefficient $K_{t}$ can be determined as a fraction of $K_{f}$, the (assumed) Krogh permeability of the fluid.

We now derive three distinct expressions for $K_{t}$ corresponding to the three possible $O_{2}$ tension profiles illustrated in Figure 1: (a) parabolic; (b) parabolic–cosh; (c) cosh.

Assume the perfusion fluid is nonflowing and biologically inert. Because no oxygen is consumed in the fluid, Equation (Equation 4) for $O_{2}$ diffusion within the fluid reads
$${\left(\frac{\partial^{2}P}{\partial x^{2}}\right)}_{f}=0\;\Rightarrow \;{\left(\frac{\partial P}{\partial x}\right)}_{f}=c_{1}\;\Rightarrow \;P_{f}\left(x\right)=c_{1}x+c_{2}$$
with $c_{1}$, $c_{2}$ being constants. Thus pressure varies linearly with depth within the nonflowing fluid layer. Let the stationary layer have thickness Δx, then from definition (Equation 5), the $O_{2}$ flux across the fluid layer will be,
(19)$$\begin{array}{c} q_{f}\;=\;K_{f}\;{\left(\frac{\partial P}{\partial x}\right)}_{f}\;\approx \;K_{f}\;{\left(\frac{\Delta P}{\Delta x}\right)}_{f} \end{array}$$
where ΔP is the pressure change across fluid layer Δx.

**(a)** 
**Parabolic profile**


Define coefficient $a\equiv (Q_{0}/2K)$ in Equation ([Disp-formula FD8a-ijms-24-06450]). Then, $O_{2}$ tension in tissue is
(20)$$\begin{array}{cc} P_{t}\left(x\right)\; & =\;P_{s}-a\left(L^{2}-x^{2}\right) \end{array}$$
giving a pressure gradient at the x=L boundary,
(21)$$\begin{array}{cc} {\left.\left(\frac{\partial P_{t}}{\partial x}\right)\right|}_{x=L}\; & =\;2aL \end{array}$$
so the incoming $O_{2}$ flux at the fluid–tissue interface will be
(22)$$\begin{array}{cc} q_{t}\left(L\right)\; & =\;K_{t}{\left.\left(\frac{\partial P_{t}}{\partial x}\right)\right|}_{x=L}\;=\;2aLK_{t}. \end{array}$$
Equating fluid flux (Equation 19) with tissue-boundary flux (Equation 22), we obtain the desired expression for Krogh permeability in tissue,
(23)$$\begin{array}{c} K_{t}\;=\;K_{f}\;{\left(\frac{\Delta P}{\Delta x}\right)}_{f}\cdot \frac{1}{2aL} \end{array}$$
Note that is a particular case of the general form which applies for all tissue profiles,
(24)$$\begin{array}{c} K_{t}\;=\;K_{f}\;{\left(\frac{\Delta P}{\Delta x}\right)}_{f}\;/\;\;{\left.\left(\frac{\partial P_{t}}{\partial x}\right)\right|}_{x=L} \end{array}$$

**(b)** 
**Parabolic–cosh mixed profile**


Making the following substitutions in (10b) in favour of coefficient *a*,
$$\frac{Q_{0}}{2K_{t}}=a,\;\alpha =\sqrt{\frac{Q_{0}}{P^{*}K_{t}}}=\sqrt{\frac{2a}{P^{*}}},\;\alpha P^{*}=\sqrt{2aP^{*}}$$
we obtain the pressure gradient at the x=L boundary,
(25)$$\begin{array}{cc} {\left.\left(\frac{\partial P_{t}}{\partial x}\right)\right|}_{x=L}\; & =\;\sqrt{2aP^{*}}\;\tanh\left(\sqrt{\frac{2a}{P^{*}}}\;\delta \right)\;+\;2a\left(L-\delta \right) \end{array}$$
Substituting (Equation 25) in (Equation 24) gives the tissue Krogh coefficient for the mixed pressure profile (hyperbolic-cosine centre with parabolic wings). Note that in the limit δ→0, the paracosh boundary gradient (Equation 25) collapses to the (Equation 23) parabolic profile value of 2aL, as expected.

**(c)** 
**Cosh profile**


From ([Disp-formula FD12b-ijms-24-06450]), the pressure gradient at the fluid–tissue boundary x=L is
(26)$$\begin{array}{cc} {\left.\left(\frac{\partial P_{t}}{\partial x}\right)\right|}_{x=L}\; & =\;\alpha P_{s}\;\tanh\left(\alpha L\right)\;=\;\sqrt{\frac{2a}{P^{*}}}\;P_{s}\;\tanh\left(\sqrt{\frac{2a}{P^{*}}}\;L\right) \end{array}$$
giving tissue Krogh permeability on substitution into (Equation 24).

## 3. Materials and Methods

In this section, we describe the experimental setup to measure pO${}_{2}$ profiles in thin slabs of mouse cortical tissue, and outline the software techniques used to analyse the profiles to extract $\left(Q/K_{t}\right)$ curvatures and $\left(K_{t}/K_{f}\right)$ Krogh ratios.

### 3.1. Tissue Preparation

The brain was rapidly dissected from adult male and female C57 mice anaesthetised with CO${}_{2}$, then submerged in ice-cold ‘Normal’ artificial cerebrospinal fluid (aCSF); see Table 3 for chemical composition. Coronal 400 μm-thick sections were sliced between +1 to −5 Bregma using a vibrotome (Campden Instruments Ltd., Sileby, Leics, UK). The slices were then immersed at room temperature in no-magnesium (no-Mg) aCSF (see Table 3, columns 4–6). All solutions were made in double-distilled water and pre-oxygenated (95% oxygen, 5% nitrogen) using an oxygen concentrator (Perfecto2, Invacare, Auckland, New Zealand) prior to use.

### 3.2. Data Recording

At least 60 min after preparing the tissue, the slices were transferred one at a time to a submersion-style perfusion bath based on a design by Thomas [24] (see Figure 5), and continuously perfused with aCSF. This design of the perfusion bath allows each hemisphere of the slice to be treated independently, but in this study both halves of the slice were exposed to identical experimental conditions. The aCSF flow rate was set at 0.5 mL/min for the repeated-profile experiments; flow values used in other experiments were [1, 2, 10] mL/min. Our expectation was that lower aCSF flow rates might favour the formation of a stationary boundary layer, but recognised that lower flows would restrict $O_{2}$ supply and could compromise tissue function.

Slices were perfused throughout with no-Mg aCSF. Removing magnesium from the aCSF unblocks NMDA receptors [25], resulting in the generation of ongoing spontaneous bursts of paroxysmal neuronal activity known as seizure-like events (SLEs) [26]; these fast voltage transients provide a convenient and sensitive indicator of tissue viability, and allow correlation of neurophysiological activity with tissue-oxygen profile characteristics.

SLEs were detected using a 75 μm diameter Ag/AgCl wire electrode inserted into the 400 μm-thick tissue slice, and measured as an extracellular local-field potential difference developed between active tissue and a common-ground electrode (Ag/AgCl disc) placed some distance from the slice in the perfusion bath. The analog signal was amplified (1000×) and filtered (bandpass 1–100 Hz, notch filter at 50 Hz) (Model 3000 differential amplifier, A-M Systems, Sequim, WA, USA), then digitised at 1000 s${}^{-1}$ (PowerLab, ADInstruments, Bella Vista, NSW, Australia) and recorded using LabChart 7 (ADInstruments, Australia).

**Figure 5 ijms-24-06450-f005:**
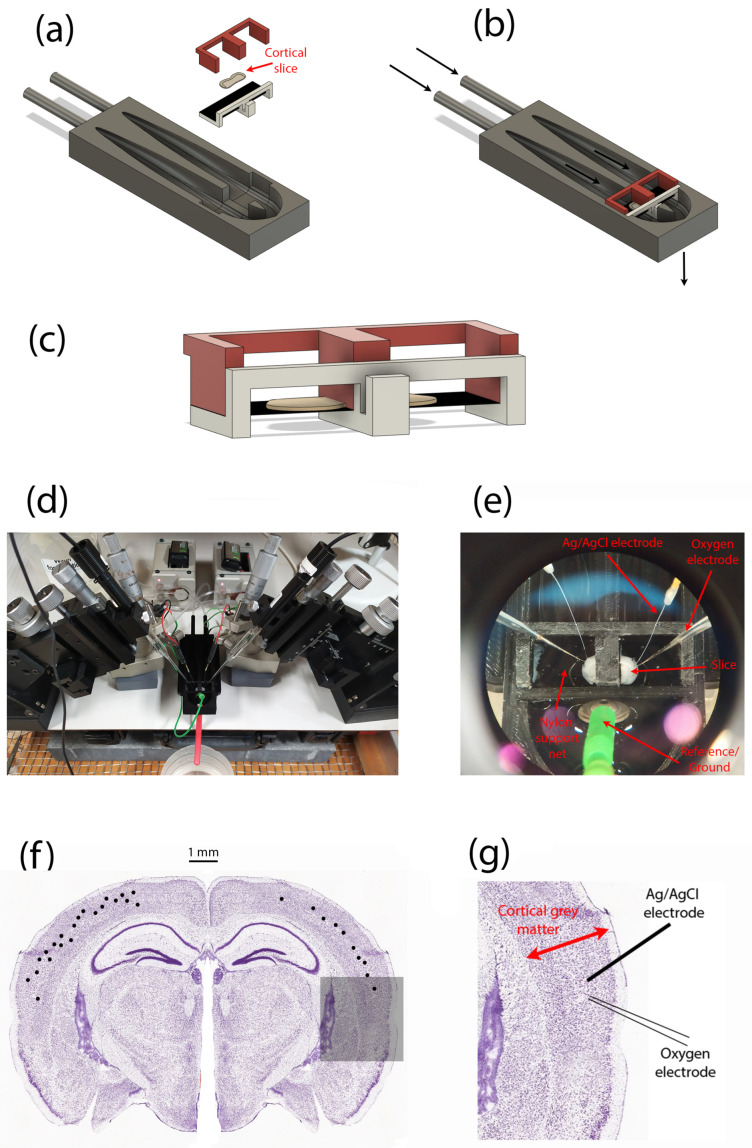
Perfusion bath and experimental setup for measuring oxygen-pressure profiles within a thin slice of mouse cortical tissue. (**a**) Exploded CAD model showing the separate components of the dual-compartment slice perfusion apparatus. (**b**) Assembled perfusion bath; arrows indicate direction of solution flow. (**c**) Enlarged view of the brain slice (grey ‘pancake’ shape) and its support structures; the slice sits on a nylon net (shown as a black-shaded horizontal surface). (**d**) Overview of experimental setup showing left and right pairs of micromanipulators, local-field potential (LFP) electrodes, oxygen probes. (**e**) Zoomed view of in situ cortical slice sitting on nylon net, and probed by LFP (Ag/AgCl wires) and oxygen (glass) electrodes. (**f**) Distribution of recording locations (black dots) within the slice cerebral cortex. For illustrative purposes, locations are shown on a single slice (but slices anterior to the one shown were used in some cases). Repeat profiles captured from some locations are not differentiated. (**g**) Representative arrangement of LFP electrode and oxygen probe for a single recording location, expanded from the shadowed region in (**f**).

Oxygen partial pressures (pO${}_{2}$) were measured using a Clark-style oxygen electrode (50 μm tip diameter, Unisense Ltd., Aarhus, Denmark) inserted into the tissue slice. Prior to experiments, the oxygen electrode was polarised to ensure a stable output signal, then two-point calibrated using (a) aCSF equilibrated with room air (pO${}_{2}$ = 160 mmHg), and (b) a solution of 0.1-M sodium ascorbate (zero-point). The pO${}_{2}$ data were sampled at ∼4.8 Hz using SensorTrace (v1.8, Unisense Ltd., Denmark).

### 3.3. Experimental Protocol

Once the slice was established in the perfusion bath, the Ag/AgCl wire electrode was inserted into layer III/IV of the cerebral cortex, with no specific cortical region targeted. If robust SLE activity was detected at that location (indicating viable tissue), the oxygen sensor was positioned using a precision micromanipulator (FX-117, Minitool Inc., Los Gatos, CA, USA) as close as practicable to the Ag/AgCl electrode, placing the tip of the oxygen probe in the bath fluid about 200 μm above the slice surface.

The pO${}_{2}$ vertical profile was recorded every 50 μm by lowering the oxygen probe in 50-μm steps downwards through the upper fluid layer, then through the full depth of the 400-μm tissue slab, continuing until the tip had penetrated ∼100 μm beyond the lower slice surface into the lower fluid layer. It was essential to include $O_{2}$ soundings in the upper and lower fluid layers for two reasons. First, because of lack of visual contrast, it was not possible to accurately discern when the probe entered or exited the tissue, so the locations of the slice centre (x=0) and slice boundaries (x=±L) could only be determined later, during the curve-fitting stage, by assuming that the curve minimum marked the (zero-flux) centre of symmetry. Second, we were seeking evidence of a nonflowing boundary layer in order to apply the flux-conservation argument of Equation (Equation 19); this would manifest as an abrupt change in profile gradient on crossing the tissue–fluid boundary as the *P* vs. *x* profile curve transitions from parabolic to linear.

Some profiles exhibited a smooth tissue–fluid boundary crossing similar to those shown in Figure 3, so were not suitable for estimation of tissue Krogh coefficient. In contrast, Figure 6a,b,d display a clear gradient discontinuity at the lower tissue–fluid interface, but no gradient break at the upper interface. We attribute this profile asymmetry to a design characteristic of the slice perfusion system: the finely-woven nylon netting on which the slice sits (see Figure 5c,e) seems to support the formation a locally static fluid layer immediately adjacent to the lower surface of the slice, without impeding the bulk flow of solution below the net. If the fluid is static (i.e., no advection), then oxygen transport through the fluid layer adjacent to the slice is via diffusion only; therefore, diffusive flux through the layer should match that entering the tissue, so the Equation (Equation 19) assumption of flux conservation should be applicable here.

The fact that the probe detects a lower-surface stagnant layer in many—but not all—profiles suggests that the emergent location of the probe tip relative to the weave structure of the netting may be significant, but this aspect remains unresolved at present.

### 3.4. Numerical Methods and Curve Fitting

Each pressure profile consists of a vector of ∼20 oxygen tension readings [mmHg] recorded at 50-μm intervals during a micromanipulator-controlled vertical descent starting in the fluid above the slice, then stepping down through the fluid into the tissue slice, through the slice, then continuing for several steps beyond the tissue into the fluid beneath. Tissue thickness is 2L=400
μm, so either 8 or 9 samples will lie within the tissue. The precise position of the oxygen probe tip relative to the tissue boundary is not easily visualised, so the metabolic centre of the slice is determined mathematically by fitting a parabola to the profile: the dP/dx=0 pressure minimum locates the zero-flux reference depth x=0.

**Figure 6 ijms-24-06450-f006:**
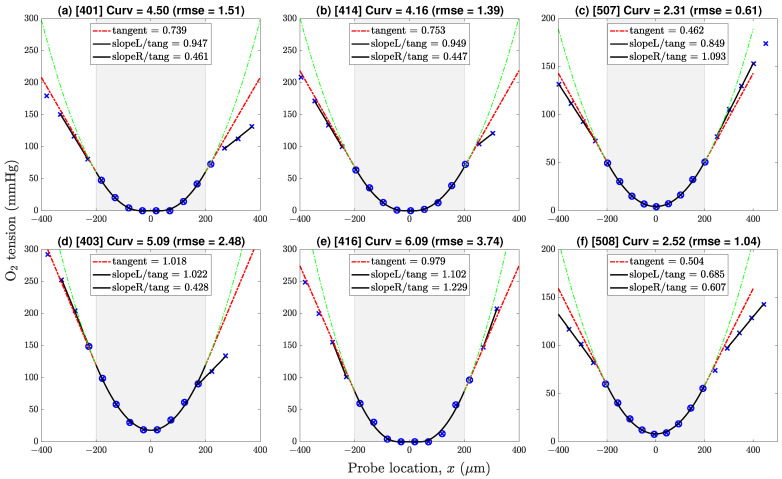
Representative $O_{2}$ pressure profiles drawn from datasets v4 (first two columns) and v5 (third column). Profiles (**a**,**b**,**d**,**f**) show a gradient discontinuity at the lower boundary; profiles (**c**,**e**) do not. Negative/positive locations correspond to points lying above/below the x=0 line of symmetry. Key: × = sample values; ⊗ = samples selected for 9-point in-tissue curve-fit (parabola or paracosh); dashed-green = extrapolation of parabola/paracosh curve into fluid layers above (left) and below (right) the tissue–fluid interface; dashed-red = tangent to curve at boundary with slope ${\left(\partial P/\partial x\right)|}_{x=L}$ [mmHg/μm]; solid-black linear segments identify linear pressure trends in proximal fluid layer; ‘Curv’ = $10^{3}$ × curvature = $10^{3}$ × (2a) [mmHg/μm${}^{2}$]. Gradient ratios ‘slopeL/tang’, ‘slopeR/tang’ give estimates for above-slice, below-slice $\left(K_{t}/K_{f}\right)$ Krogh ratios; ratios exceeding cutoff value 0.725 are rejected.

**Case (a):** If the profile shape is accurately modelled as parabolic, then Equation (Equation 20) applies, and the oxygen pressure gradient in tissue is given by
$$\frac{\partial P_{t}}{\partial x}\;=\;2ax\;\to \;2aL\;\mathrm{at}\;\mathrm{the}\;x=L\;\mathrm{boundary}$$

**Case (b):** If the pressure curve is ‘flat-bottomed’, then Equations (9) and (10) apply, indicating that the profile is a ‘paracosh’ mixed case with a hyperbolic-cosine (cosh) core for (−δ≤x≤δ), and parabolic wings for (|x|>δ). Paracosh fitting proceeds by iterating curvature $(Q/K_{t})=2a$ and critical pressure $P^{*}$ to maximise agreement between the paracosh curve and the subset of $\left(x_{i},P_{i}\right)$ data points lying within the tissue boundaries. Paracosh optimisation makes use of Matlab function fminsearchbnd [D’Errico (2022), www.mathworks.com/matlabcentral/fileexchange/8277-fminsearchbnd-fminsearchcon, (accessed 29 June 2022)] which adds bounded constraints to the standard fminsearch optimiser; in our case, we require that both 2a and $P^{*}$ be non-negative. The critical depth $\delta =\delta (P_{s},P^{*},2a)$ is obtained via a separate iteration on Equation (Equation 11) with surface pressure $P_{s}$ fixed by linear interpolation of the $\left(x_{i},P_{i}\right)$ data pairs bracketing the x=±L boundaries. Once the paracosh curve parameters $\left(2a,P^{*},\delta \right)$ have been established, the pressure gradient at the x=L boundary is given by Equation (Equation 25).

**Case (c):** The cosh-only profile ($P_{s}<P^{*}$) was never encountered in any of our oxygen tension soundings; nevertheless, the case-(b) curve-fitting algorithm should work equally well here.

If a clear gradient break could be identified at the tissue–fluid boundary, a straight line was fitted to the linear pressure trend in the fluid adjacent to the slice edge. The tissue–fluid Krogh ratio $\left(K_{t}/K_{f}\right)$ was then obtained using the flux conservation expression of Equation (Equation 24).

## 4. Results

We analysed five distinct pO${}_{2}$ profile datasets [labelled v1 to v5] but discarded the first two because of oxygen-probe calibration issues. The remaining datasets are:

**Dataset v3** (10 mL/min): Recorded from 19 cortical locations from 4 slices (1 animal); n=19 profiles

**Dataset v4** (1 and 2 mL/min): Recorded from 8 locations from 4 slices (1 animal), repeated at 1 and 2 mL/min for each location; n=16 profiles

**Dataset v5** (0.5 mL/min): Recorded from 6 cortical locations from 2 slices (1 animal), each profile repeated once, giving 6 profile pairs; n=12 profiles

This gave a total of 47 oxygen tension profiles as summarised in Table A1. Most of the curves (31 of 47) were well-fitted with a simple parabola; the remainder (16 of 47) were fitted with a mixed paracosh function (identified with ‘Flat = 1’ table entry); none of the profiles exhibited a purely cosh shape. The proportion of flattened curves decreased as aCSF flow rate was reduced: [42%, 31%, 25%] for [v3, v4, v5], respectively. Provided that each curve was correctly classified as parabolic or paracosh, we found no discernible difference between $K_{t}$ estimates derived from the parabolic set compared with the paracosh set.

### 4.1. Oxygen Tension Profiles in Fluid and Tissue

Figure 6 shows six representative profiles with pressure gradient discontinuities evident at neither, one, or both tissue–fluid boundaries at x=±L, slice half-thickness being L=200
μm. If the adjacent fluid exhibited a linear P(x) trend, we took this as evidence of a local nonflowing fluid layer which should permit estimation of the ($K_{t}/K_{f}$) Krogh ratio via conservation of $O_{2}$ flux across the interface (see Equation (Equation 24)).

For each P(x) pressure profile, Krogh ratio retrievals proceed via two independent curve-fitting steps. First, the optimal value of profile curvature $(Q/K_{t})\equiv 2a$ is determined by minimising the rms difference between measured $\left(x_{i},P_{i}\right)$ data points and iterated $\left(x_{i},P\left(x_{i}\right)\right)$ parabola/paracosh predictions for the eight or nine locations $x_{i}$ lying within the −L≤x≤L tissue boundaries. The resulting pressure gradient at the boundary ${\left(\partial P/\partial x\right)|}_{x=L}$ is then calculated using Equation (Equation 21) (parabola) or Equation (Equation 25) (paracosh): see red ‘tangent’ extrapolations in Figure 6.

Second, the pressure values in the proximal fluid above and below the slice are inspected for linear trends ΔP/Δx, whose slope is markedly *lower* from the tissue tangent extrapolations; a clear gradient break suggests a locally stationary fluid layer. For our experimental setup, these gradient breaks were common at the lower interface (at x=L), but rarely occurred at the top interface (x=−L), except at the lowest perfusion rates. We attribute this top/bottom—left/right on the graphs—asymmetry to the presence of the fine nylon mesh that supports the underside of the slice (see Figure 5); evidently, the weave of the net can create a ‘shadow zone’ that shields the slice from longitudinal advective currents.

If the fluid pressure gradient was similar to, or larger than, the extrapolated tissue gradient (i.e., if $(K_{t}/K_{f})≳1$), then the candidate ratio was immediately rejected (no stationary layer, therefore flux conservation assumption is invalid). However, a more stringent acceptance criterion is needed since some apparently stationary cases may be ‘contaminated’ by weak residual advective flows. After inspecting scatter plots of Krogh ratio vs. curvature (see Figure 7 and Figure 8a), and Krogh ratio histograms for $(K_{t}/K_{f})<1$ (Figure 9), we set the ratio cutoff at
$${\left(K_{t}/K_{f}\right)}^{\max}\;=\;0.725$$

This selection was made on the basis that candidate Krogh ratios evidently fall into two clusters:candidate ratios larger than unity are biologically disallowed (oxygen permeability in tissue cannot be greater than permeability in fluid);candidate ratios larger than 0.73 appear to form part of the $K_{t}/K_{f}>1.0$ ‘disallowed’ cluster (e.g., see panel (a) of Figure 7);the aggregated histogram ratios of Figure 9 suggest a clear break between ‘allowed’ and ‘disallowed’ clusters if we set the cutoff at $K_{t}/K_{f}=0.725$.

### 4.2. Scatter Plots of Krogh Ratio vs. Curvature

Individual scatter distributions for $\left(K_{t}/K_{f}\right)$ candidate Krogh ratios vs. profile curvature are displayed as separate panels in Figure 7, then aggregated into a unified cluster-graph in Figure 8a; Figure 8b shows the apparently sigmoidal dependence of profile curvature on perfusion flow rate. The scatterplots of Figure 7 are segregated by interface (upper/lower) and by dataset (v3/v4/v5). Perfusion rates decrease from left to right: 10 mL/min (v3), 1 or 2 mL/min (v4), 0.5 mL/min (v5). We see that, on average, lower flow rates are associated with lower curvatures (‘flatter’ parabolic/paracosh curves). Since curvature = $2a=(Q/K_{t})$, then $O_{2}$ consumption rate also scales down as $O_{2}$ supply becomes more restricted (assuming $K_{t}$ can be taken as a nominal constant). In contrast, flow rate appears to have no influence on *plausible* Krogh ratios (i.e., those that fall below the proposed 0.725 cutoff). These contrary sensitivities to fluid $O_{2}$ transport rates are summarised in Table 4.

The distribution of candidate Krogh ratios within a given dataset (v3/v4/v5) appears qualitatively rather noisy and erratic, unlike the curvature distributions. This is particularly evident in the paired observations in v4 and v5. For v4, each slice location was profiled twice, first at perfusion rate 1 mL/min, then at the doubled rate 2 mL/min. For every pair, doubling the flow rate raised the curvature by a roughly similar proportion; however, the impact on Krogh ratio was unpredictable and scattered (e.g., see outlier pair 414/416 in Figure 7e), particularly for those candidate Krogh ratios lying in the nominated rejection zone. We attribute this increased scatter to three possible sources:Krogh ratio requires separate tissue and fluid curve fits, so variance in the ratio will be the *sum* of the individual curve-fitting variances for tissue and fluid gradients;the flux conservation argument used to derive Equation (Equation 24) is invalid if the proximal fluid layer is not stationary (hence the need to impose a Krogh ratio cutoff);formation of a local stagnant layer is not guaranteed, even within the closely woven structure of the nylon net that supports the slice.

We note that reducing the flow rate increases the proportion of candidate Krogh ratios that lie within our nominal acceptance range (i.e., $(K_{t}/K_{f})<0.725$), particularly for the lower interface (see bottom row of Figure 7). Evidently, stagnant layer formation is more probable at low flow rates.

**Table 4 ijms-24-06450-t004:** Sensitivity of profile curvature and Krogh ratio to variations in aCSF flow rate. Column headings identify the datasets used for computing statistics; the v4-dataset is partitioned into its high- and low-flow subsets. Perfusion rates decrease from left to right across the columns. Krogh statistics summarise the x=L (lower interface) retrievals, but note that candidate Krogh ratios that exceed the ${\left(K_{f}/K_{f}\right)}^{\max}=0.725$ cutoff have been excluded (see Table A1).

	v3	v4 (hi)	v4 (lo)	v5	All
Flow (mL/min)	10.0	2.0	1.0	0.5	
**Curvature**, (2a)					
$10^{3}$ ×mean	6.55	5.02	4.23	3.14	
$10^{3}$ ×stdev	1.38	0.85	0.80	0.90	
*N*	19	8	8	12	
**Krogh ratio**, $\left(K_{t}/K_{f}\right)$					
mean	0.553	0.595	0.538	0.568	0.562
stdev	0.083	0.140	0.082	0.072	0.088
*N*	11	5	5	9	30

**Figure 7 ijms-24-06450-f007:**
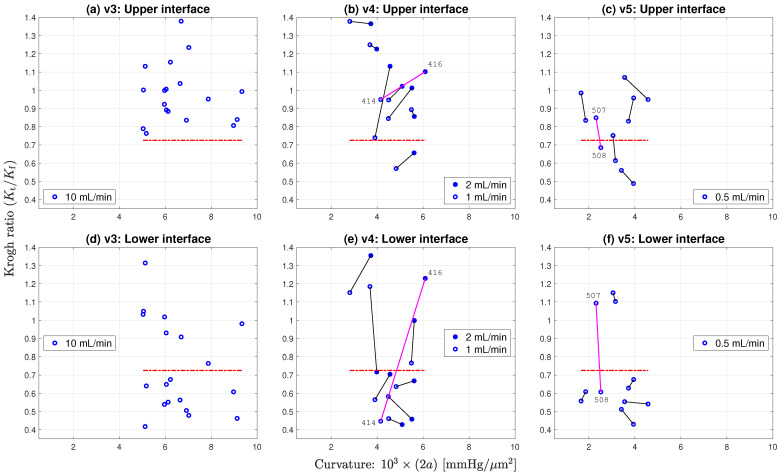
Distribution of Krogh ratios as a function of curvature of fitted parabolic/paracosh function. Results are clustered by dataset (three columns: v3/v4/v5) and interface (two rows: upper/lower). Dashed-red horizontal marks the selected cut-off between accepted (below red line) and rejected (above line) Krogh ratios. Scanning from left to right, lower aCSF flow rates are generally associated with reduced curvature values, implying increasingly constrained $O_{2}$ consumption. Linked pairs show repeated sampling at the same location. For v4, flow rate was set at 1 (open circles) or 2 mL/min (filled circles); for v5, flow rate was fixed at 0.5 mL/min. Outlier pairs 414/416 (v4) and 507/508 (v5) have very discrepant Krogh ratio estimates at the lower interface, possibly due to mechanical disturbance of the slice during withdrawal of $O_{2}$ probe prior to repeat sounding. See Figure 6 and Table A1.

**Figure 8 ijms-24-06450-f008:**
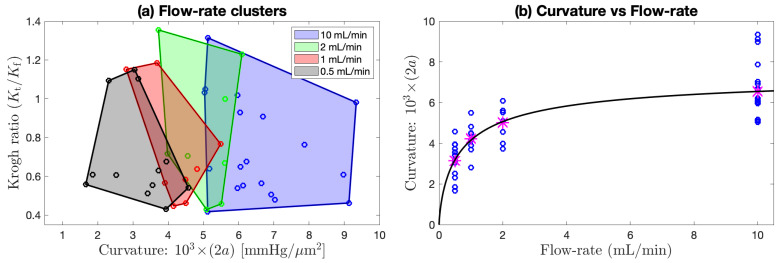
Flow rate clustering and curvature dependence aggregated across (lower interface) of Figure 7d–f. (**a**) Aggregated scatterplot of candidate Krogh ratio $\left(K_{t}/K_{f}\right)$ vs. curvature of fitted parabolic/paracosh function, clustered by flow rate [10, 2, 1, or 0.5] mL/min, as indicated by shaded convex-hull polygons (computed via Matlab function convhull). Qualitatively, the polygon centroids move to the left as flow rate decreases, implying that curvature decreases (profiles become flatter) as perfusion flow rate is reduced. This trend is made quantitative in (**b**) with a sigmoid fit to the Table 4 curvature means (magenta asterisks) at each flow rate. The fitted curve is $y=y^{\max}\cdot x^{n}/\left(K^{n}+x^{n}\right)$ with [$y^{\max}=7.5\times 10^{-3}$ mmHg/μm${}^{2}$; K=0.75 mL/min; n=0.75].

Unlike the v4 pairs, the perfusion rate for the v5 paired observations (Figure 7c,f) was maintained at a constant value (0.5 mL/min); consequently, the profile repeats typically have smaller Cartesian separation (the links are smaller) than is the case for v4, and the average link length can be taken as an overall measure of the experimental uncertainty associated with our method. Nevertheless, we still see an outlier pair 507/508 at the lower interface with an abrupt reduction in the candidate Krogh ratio. While the 507/508 profile graphs in Figure 6c,f show consistent in-tissue curvature values, the proximal fluid environment immediately above and below the tissue changes dramatically in the ∼1 min between the 507 and 508 soundings: the later profile shows clear evidence of stationary fluid layers at the upper and lower boundaries, while no evidence was apparent in the earlier profile. Perhaps the withdrawal and reinsertion of the $O_{2}$ probe caused a subtle change in the seating of the tissue slab that favoured the formation of stagnant zones?

### 4.3. Histograms for Krogh Ratio and Profile Curvature

In Figure 9, we present histograms for Krogh ratios aggregated over the [v3, v4, v5] datasets shown in Figure 7, but restrict the analysis to $(K_{t}/K_{f})\leq 1.0$ [allowing $(K_{t}/K_{f})>1.0$ would imply that $O_{2}$ permeability in tissue *exceeds* that in fluid; this is physiologically implausible]. Krogh ratios are biased towards larger values at the upper tissue–fluid interface (Figure 9a), and smaller values at the lower interface (panel (b)); this is consistent with our observation that well-defined stagnant layers are a common occurrence at the lower boundary, but rare at the upper boundary. Setting an upper bound of ${\left(K_{t}/K_{f}\right)}^{\max}=0.725$ provides a clean separation between accepted and rejected Krogh ratio candidates.

Figure 10 histograms the profile curvatures across each of the [v3, v4, v5] datasets. It confirms the earlier observation that smaller aCSF flow rates are associated with tissue profiles that have weaker (flatter) curvature, indicating that restricting oxygen flow leads to constrained metabolic activity.

**Figure 9 ijms-24-06450-f009:**
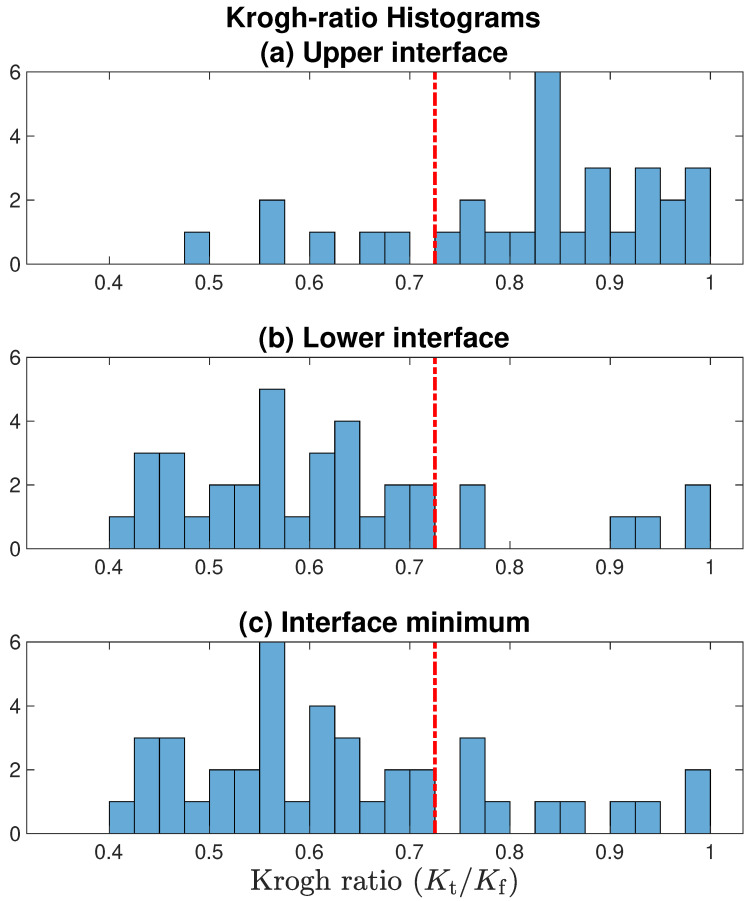
Histograms for Krogh ratios aggregated over [v3, v4, v5] datasets illustrated in Figure 7, but restricted to domain $(K_{t}/K_{f})\leq 1.0$. Red-dashed line marks the accept/reject boundary set at 0.725: only Krogh ratios below cutoff are associated with a well-defined stationary layer. Comparing panels (**a**,**b**), the lower tissue–fluid interface is more likely to form a nonflowing boundary layer. In panel (**c**), for each profile, the smaller of the [upper interface, lower interface] Krogh ratio is selected.

### 4.4. Possible Linkage between SLE Activity and Formation of Stationary Boundary Layer

We postulated that detection of a stagnant fluid layer at the tissue–fluid interface might require the maintenance of at least a minimum level of metabolic activity within the tissue. The ‘stationary fluid’ idealisation requires that—close to the tissue surface—oxygen diffusion (a molecular random walk with net $O_{2}$ motion directed towards the face of the tissue) through the fluid strongly dominates any residual advective bulk flow ($O_{2}$ in fluid moving parallel to the tissue). However, if the tissue is not drawing much oxygen from the fluid, then diffusive flux across the interface will be low, and the $O_{2}$ advective component may no longer be insignificant, meaning that the flux conservation argument fails because the local fluid layer is insufficiently ‘stationary’.

**Figure 10 ijms-24-06450-f010:**
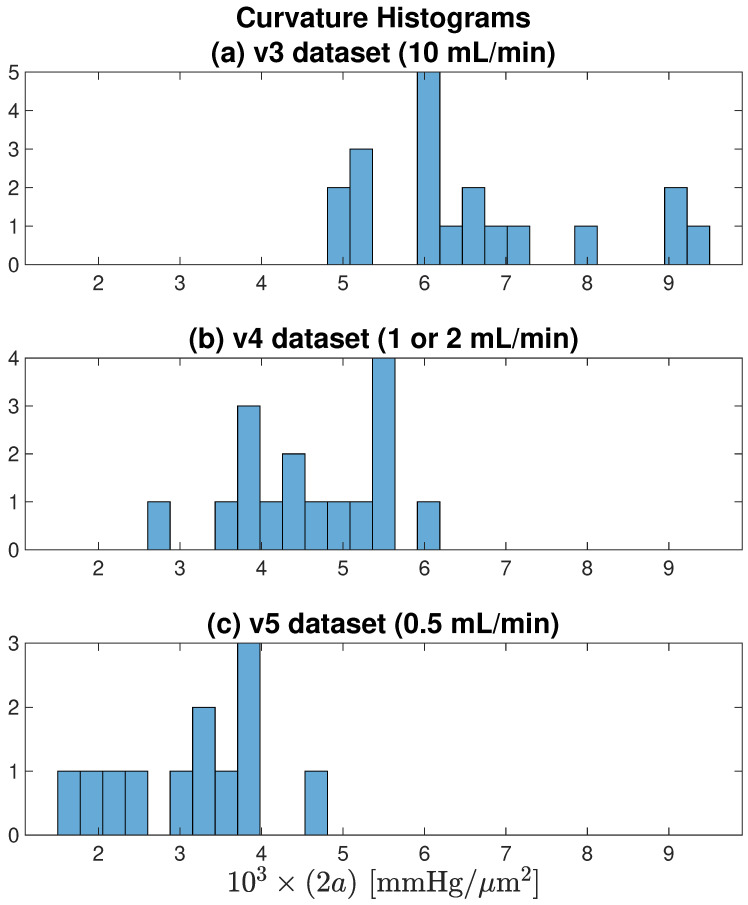
Curvature histograms for each of the [v3, v4, v5] datasets illustrated in Figure 7. As flow rate decreases from (**a**) 10 → (**b**) [1 or 2] → (**c**) 0.5 mL/min, average curvature decreases, meaning that the parabolic/paracosh curves become ‘flatter’ with shallower wings. For fixed $K_{t}$, a flatter curvature implies reduced metabolism.

We chose to test this hypothesis on the v3-database since this used the fastest aCSF flow rate (10 mL/min), thus maximising both $O_{2}$ delivery and the potential for seizure-like electrical activity. For each of the nineteen pressure soundings, we identified four consecutive seizure-like events (SLEs) that occurred around the time of profile acquisition, and computed average values for three SLE parameters: peak-to-peak amplitude [μV]; duration [s]; inter-event frequency [s${}^{-1}$].

We found that SLE amplitude (but not SLE duration or frequency) correlated *negatively* with $\left(K_{t}/K_{f}\right)$ Krogh ratio for the under-slice interface ($R^{2}=0.23$, p=0.03). That is, a tissue location generating smaller SLEs was *less* likely to display the pressure-gradient discontinuity associated with a nonflowing fluid layer at the x=L boundary. Conversely, a site generating larger SLEs was more likely to show evidence of a stationary fluid layer.

### 4.5. Estimation of Krogh Coefficient for Cortical Tissue at Room Temperature

We now have sufficient data to derive an estimate for $K_{t}$, the oxygen Krogh coefficient for mouse cortical tissue at room temperature. In Section 2.6, we computed a theoretical value for $K_{f}$, the Krogh coefficient for the no-Mg aCSF (no-magnesium artificial cerebrospinal fluid) used to supply oxygen and glucose to the 400 μm-thick slice of brain tissue (Equation 18),
(27)$$\begin{array}{c} K_{f}\;=\;\left(2.30\pm 0.04\right)\times 10^{-14}\;\mathrm{mol}/\left[m\cdot s\cdot \mathrm{Pa}\right] \end{array}$$

This value is based on that for pure water, but adjusted for the presence of dissolved salts and glucose. As displayed in Figure 4, the coefficient is weakly temperature dependent. If the aCSF temperature were to rise from 20 to 25 ${}^{{}^{\circ}}C$, $K_{f}$ would increase by ∼4% from [2.262 to 2.348] ×10${}^{-14}$ mol/[m·s·Pa]. We have chosen the middle of the range, with an uncertainty of ±2%.

From the Table 4 statistics for the tissue–fluid ratio of Krogh coefficients $\left(K_{t}/K_{f}\right)$, we learn that the Krogh ratio is completely insensitive to perfusion rate, so we are justified in combining all datasets (last column of Table 4) to give an aggregate (mean ± standard deviation) statistic,
(28)$$\begin{array}{c} \left(K_{t}/K_{f}\right)\;=\;0.562\pm 0.088\;\left(\mathrm{dimensionless};\;N=30\right) \end{array}$$

The standard deviation in the ratio is ∼16% of the mean. This uncertainty estimate is almost an order of magnitude larger than the 2% relative uncertainty in $K_{f}$ for the perfusion fluid, so the latter can be neglected as source of uncertainty in the Krogh ratio. How much of this uncertainty is due to experimental error (e.g., probe calibration and positioning, pressure measurement, in-tissue and in-fluid curve fitting, tissue movement, unsteady flow rates, degraded stagnant layer, etc.), and how much arises from the natural variability of living cortical tissue?

The [v4, v5] repeated profiles—shown as linked pairs in Figure 7e,f—allow us to apportion the relative contributions of experimental and biological sources of variation. There are eight linked pairs (four in each of [v4, v5]) that lie entirely within the $(K_{t}/K_{f})<0.725$ acceptance zone. Let (A,B) represent the (first, second) elements of the eight pairs of Krogh ratios retrieved at each repeated location. Define the biological and experimental contributions to the variance (i.e., square of the standard deviation) of the Krogh ratios as $\sigma_{\mathrm{biol}}^{2}$ and $\sigma_{\mathrm{expt}}^{2}$, respectively. Then, the following variance identities should apply,
$$\begin{array}{ccc} \mathrm{var}(A-B)\; & =\;\sigma_{\mathrm{expt}}^{2} & \;\;\;(\mathrm{biology}\;\mathrm{cancels})\\ \mathrm{var}(A+B)\; & =\;\sigma_{\mathrm{expt}}^{2}+\sigma_{\mathrm{biol}}^{2} & \;\;\;(\mathrm{both}\;\mathrm{contribute})\\ \Rightarrow \;\sigma_{\mathrm{biol}}^{2}\; & =\;\mathrm{var}(A+B)-\mathrm{var}(A-B) \end{array}$$
Our pairwise variance calculations give
$$\mathrm{var}\left(A+B\right)=0.0263,\;\mathrm{var}\left(A-B\right)=\sigma_{\mathrm{expt}}^{2}=0.0069,\;\Rightarrow \;\sigma_{\mathrm{biol}}^{2}\;=\;0.0194$$
implying that $[\sigma_{\mathrm{biol}}^{2}/\left(\sigma_{\mathrm{expt}}^{2}+\sigma_{\mathrm{biol}}^{2}\right)]=74\%$ of the variance in our Krogh ratio determinations is biological in origin, and the remaining 26% is attributable to experimental uncertainty.

Finally, we compute the room temperature Krogh coefficient for mouse cortical tissue by taking the product of (Equation 27) and (Equation 28),
(29)$$\begin{array}{c} K_{t}\;=\;\left(1.29\pm 0.21\right)\times 10^{-14}\;\mathrm{mol}/\left[m\cdot s\cdot \mathrm{Pa}\right] \end{array}$$
where we have carried forward the 16% uncertainty from (Equation 28).

## 5. Discussion

In this paper we have sought to provide a theoretical and experimental basis for determining oxygen consumption in thin slices of mouse brain tissue. Oxygen consumption within metabolically active tissue can be deduced from the curvature of pO${}_{2}$ oxygen profiles using Fick’s law of diffusion, so long as the oxygen diffusion and solubility coefficients of tissue are known. Because oxygen solubility in tissue is difficult to measure, it is standard practice to work with the Krogh coefficient $K_{t}$, a lumped measure of oxygen permeability given by the product of diffusion and solubility coefficients for oxygen. Early work by Buerk & Saidel [27] identified the Michaelis–Menton kinetics model as the most accurate description of pO${}_{2}$ gradients in tissue slices; its piecewise-linear approximation (see Equation (Equation 6)) provides the basis of the analytical solution published by Ivanova & Simeonov [10] and extended here. We followed Ganfield et al. [8] in assuming the existence of a narrow but stationary (i.e., non-flowing) boundary layer of fluid in close proximity to the tissue surface, then invoking conservation of oxygen flux across the fluid/tissue interface in order to deduce a value for $K_{t}/K_{f}$, the dimensionless ratio of tissue–fluid Krogh coefficients. By construction, the perfusion fluid is a passive, non-biologically active medium, so the fluid Krogh coefficient $K_{f}$ can be calculated from well-established interpolation formulas that are functions of both water temperature and saline concentration; this then allows tissue Krogh coefficient $K_{t}$ to be determined.

There are alternative ways to measure the tissue Krogh coefficient experimentally; however, these are generally not appropriate for soft, delicate tissue such as brain. For example, the original gaseous diffusion method described by Krogh (1919) [16] involved separating two chambers with a stretched membrane, allowing measurement of gas diffusion through the membrane from one chamber to the other. A similar direct measurement method was applied by Sasaki et al. [14], using microscopy techniques to measure the flux of oxygen across thin (10–20 μm) arteriolar walls. Such methods are not easily applied to a slice of brain tissue since it is very easily damaged by physical manipulation.

It is informative to compare our value for tissue Krogh coefficient with those published in the literature. This requires appropriate unit conversions from SI to the various alternative unit systems in use. We select two of the more common metric choices:
(mL O_2_)/(cm·min·atm): Ganfield et al. [8](mmol O_2_)/(cm·min·mmHg): Ivanova & Simeonov [10]
and list the unit-remapped values for our estimate for $K_{t}$,
$$\begin{array}{cc} K_{t}\; & =\;\left(1.29\pm 0.21\right)\times 10^{-14}\;\mathrm{mol}/\left[m\cdot s\cdot \mathrm{Pa}\right]\\ \; & =\;\left(1.76\pm 0.29\right)\times 10^{-5}\;\mathrm{mL}/\left[\mathrm{cm}\cdot \min\cdot \mathrm{atm}\right]\\ \; & =\;\left(1.03\pm 0.17\right)\times 10^{-9}\;\mathrm{mmol}/\left[\mathrm{cm}\cdot \min\cdot \mathrm{mmHg}\right] \end{array}$$

Using the second set of units (mL/[cm·min·atm]), Ganfield et al. (1970) derive a Krogh coefficient for cat cortex of $1.29\times 10^{-5}$, about 27% lower than our $1.76\times 10^{-5}$ estimate. Their result is unexpectedly low, given that they were working at 37 ${}^{{}^{\circ}}C$, while our value was derived at room temperature.

Ivanova & Simeonov (2012) tabulate a range of Krogh coefficients at 37 ${}^{{}^{\circ}}C$ for a variety of different tissues drawn from the work of several authors. Using the third set of units (mmol/[cm·min·mmHg]), their quoted values for $10^{9}\times K_{t}$ covered the range0.59 (kidney),  0.65 (liver),  1.35 (brain),  1.44 (heart),
so our value of 1.03 × 10${}^{-9}$ for mouse cortex is certainly plausible, given the temperature difference.

We have focused on calculating tissue/fluid Krogh ratio $\left(K_{t}/K_{f}\right)$ in the first instance. The advantage of working with a dimensionless ratio is that it should aid direct comparison between studies since it eliminates the need for non-SI unit conversions. Although the biological significance of expressing Krogh permeability as a ratio is uncertain, it is plausible that the $K_{t}/K_{f}$ ratio may be less sensitive to temperature than either component. If so, this would be advantageous when attempting to compare the findings of different research groups working with a range of perfusion temperatures. This idea remains to be tested experimentally.

Our method for computing the Krogh ratio is dependent of the formation of a stationary boundary layer at the tissue–fluid interface. As discussed previously, a nonflowing fluid layer was often detected at the lower interface, but rarely at the upper interface. This asymmetrical behaviour is probably a favourable artifact created by the support netting on which the slice sits: it seems that the tight weave of the netting can provide shielding from the bulk advective flow. This likely explains some of the variability in our $K_{t}$ estimate, because precise positioning of the oxygen electrode within this stationary layer could not be guaranteed from one recording location to the next.

Importantly, when (repeat) profiles were collected from the *same* location, the variance in the $K_{t}/K_{f}$ ratio was substantially lower (contributing only 26% of the total variation), indicating that the majority of the variability was of biological origin. This hints at the possibility that the oxygen permeability characteristics of brain tissue are not uniform across and between slices, with potential influences from variation in cortical layer structure, variation in regional cortical anatomy, and spatial differences in tissue viability. In these experiments, we did not attempt to control for specific cortical anatomical location or for non-uniformity in slice viability.

In summary, we have found that the Ivanova & Simeonov diffusion–consumption model provides an excellent description of oxygen-tension distribution within a thin slice of active tissue. We have extended the model to include the effects of a stationary fluid layer at the boundary, and have shown how to compute the ratio of tissue and fluid Krogh coefficients via a flux conservation argument. Being dimensionless, the Krogh ratio allows unambiguous and direct comparisons between studies by different researchers, since it obviates the need for unit conversions. The mapping to dimensioned Krogh coefficients can be delayed until the choice of units for the fluid Krogh coefficient has been made.

## Figures and Tables

**Figure 1 ijms-24-06450-f001:**
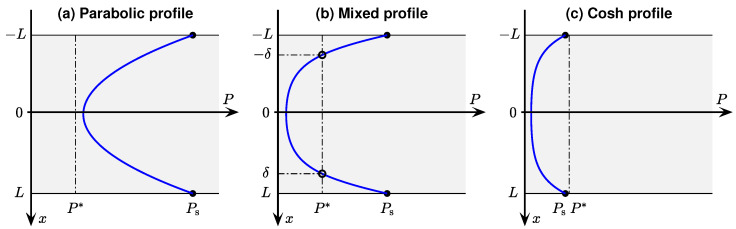
Representative P(x) solutions of Ivanova & Simeonov diffusion–consumption Equations (Equation 4) and (Equation 5), assuming a piecewise linear model (Equation 6) for consumption rate (see blue curve in Figure 2). Shading represents thin slab of tissue which extends from −L to *L*. Here, $P_{s}=$ surface pressure; $P^{*}=$ critical pressure below which oxygen consumption is restricted; δ= depth at which pressure reaches critical value. (**a**) If minimum tension exceeds $P^{*}$, pressure profile is a simple parabolic function of depth. (**b**) If the pressure profile crosses the $P^{*}$ boundary, the central portion forms a flattened hyperbolic-cosine (cosh) ‘basin’ that smoothly merges with parabolic ‘wings’ for $P>P^{*}$. (**c**) If the surface tension falls below the critical value, the cosh basin extends to the tissue boundaries.

**Figure 3 ijms-24-06450-f003:**
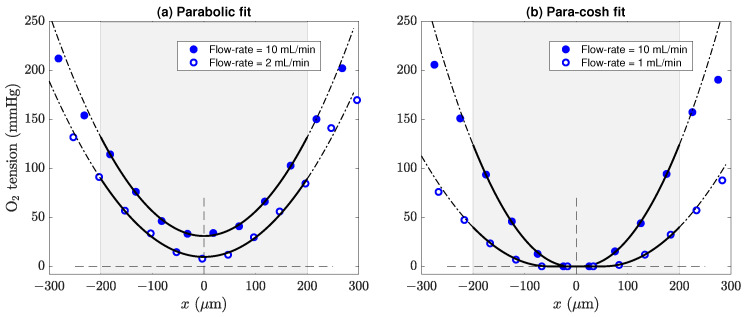
Parabolic and para–cosh curve fits for four representative tension vs. depth profiles measured from healthy slices of mouse brain tissue at room temperature (∼20 ${}^{{}^{\circ}}$C). The grey shading indicates the 400-μm extent of the slice for −200≤x/μm≤200, with x=0
μm marking the central plane of symmetry of the slice. (**a**) Parabolic profiles showing no restriction of metabolic rate. (**b**) Insufficient oxygen tension leads to flattened profiles with a cosh-modulated central basin joined to parabolic wings to the left and right.

**Figure 4 ijms-24-06450-f004:**
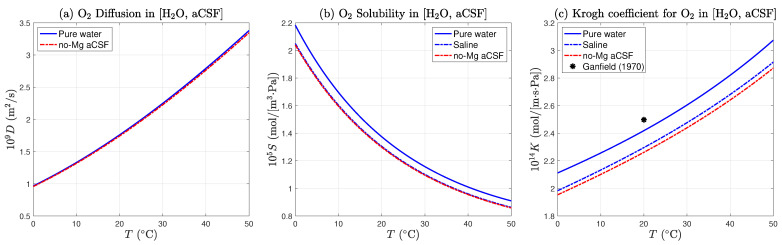
Temperature dependence of oxygen diffusion, solubility, and permeability in pure water, and in no-Mg artificial cerebral spinal fluid (aCSF). (**a**) Oxygen diffusion in aCSF (dashed-red trace) is depressed by ∼1% relative to pure water (solid blue trace) by small viscosity increase caused by presence of 0.02 mol/L glucose. (**b**) Oxygen solubility in aCSF is reduced by ∼5% by ‘salting out’ effect of chloride salts (dashed-blue trace) with chlorinity $[m_{\mathrm{Cl}}]=4.904$ g/kg; and by a further ∼0.5% by glucose effect. (**c**) Oxygen Krogh coefficient in fluid is given by the product of diffusion and solubility, K=DS. Glucose effect is assumed to be cumulative, resulting in a ∼1.5% depression below the saline curve. Asterisk (*) marks the value ($K^{*}=2.5\times 10^{-14}$ mol/(m·s·Pa)) used by Ganfield et. al. (1970) [8] for Krogh coefficient for fluid at 20 ${}^{{}^{\circ}}C$.

**Table 1 ijms-24-06450-t001:** Selection of non-SI units for oxygen Krogh coefficient appearing in the physiology literature. The numerator represents quantity of gas, e.g., (mmol O${}_{2}$), (mL O${}_{2}$), (cm${}^{3}$ O${}_{2}$), etc. Bracketed rows of the table indicate equivalent units.

	Unit	Reference	
1	$\frac{\mathrm{mol}}{m\cdot s\cdot \mathrm{Pa}}$	SI	
2	$\frac{\mathrm{mmol}}{\mathrm{cm}\cdot \min\cdot \mathrm{mmHg}}$	Ivanova & Simeonov (2012) [10]	
3	$\frac{\mathrm{mmol}}{\mathrm{cm}\cdot \min\cdot \mathrm{torr}}$	Kawashiro & Scheid (1976) [12]	
4	$\frac{\mathrm{nM}\cdot {\mathrm{mm}}^{3}}{\mathrm{mm}\cdot s\cdot \mathrm{mmHg}}$	van der Laarse et al. (2005) [13]	
5	$\frac{\mathrm{mL}}{\mathrm{cm}\cdot \min\cdot \mathrm{atm}}$	Ganfield et al. (1970) [8]	
6	$\frac{\mathrm{cm}{}^{3}}{\mathrm{cm}\cdot \min\cdot \mathrm{atm}}$	Chen & Liew (1975) [11]; Poole et al. (2020) [15]	
7	$\frac{\mathrm{mL}}{\mathrm{cm}\cdot s\cdot \mathrm{mmHg}}$	Sasaki et al. (2012) [14]	

**Table 2 ijms-24-06450-t002:** Ideal-gas molar volumes (L/mol) for STP and NTP temperature and pressure definitions. Bracketed value in first row is the actual (non-ideal) molar volume for O${}_{2}$ at STP (pre-1982).

Standard	Temperature, *T*	Pressure, *p*	Molar Volume, $V_{m}$
STP (IUPAC to 1982)	273.15 K (0 ${}^{{}^{\circ}}C$)	1 atm = 101.325 kPa	22.414 (22.392)
STP (IUPAC after 1982)	273.15 K (0 ${}^{{}^{\circ}}C$)	100 kPa	22.711
NTP (NIST)	293.15 K (20 ${}^{{}^{\circ}}C$)	1 atm	24.055

## Data Availability

The Matlab paracosh curve-fitting codes plus associated sample pO${}_{2}$ depth profiles will be supplied on request to the corresponding author (D.A.S.-R.).

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
