# Peer review of "Determination of Krogh Coefficient for Oxygen Consumption Measurement from Thin Slices of Rodent Cortical Tissue Using a Fick’s Law Model of Diffusion"

_ijms, 2023, doi:10.3390/ijms24076450_

Round 1

Reviewer 1 Report

The authors applied Fick’s-law Model of Diffusion to estimate oxygen partial pressure curve and finally deduce Krogh Coefficient for tissue slices. The authors also conducted experiments and obtained plausible results to demonstrate their theory. The limitations were discussed sufficiently. However, there are some major concerns that need to be carefully addressed.  

1. The Abstract did not directly explain how the Krogh Coefficient for tissues would help assess metabolic health of tissue slices. The text from "To assess metabolic health of the slice,.."  (line 7-17) introduced all that the authors had done in this paper, but did not tell which one is the metric to quantify the metabolic health.  Oxygen partial pressure?  Or the rate of oxygen consumption (Q)?  Krogh coefficient (Kt)? How will the metric(s) affect the health of tissues? Even though the introduction section gave more details, I still do not know the answer.

2. There is little background of Krogh Coefficient (Kt). How did other literature obtain Kt? How challenge to measure Kt? Any peer methods for comparison? What did other literature use Kt for? Are there any other applications of Kt besides quantifying tissue health? The authors should add more introduction of Kt.

3. If the paper only needs to determine Krogh Coefficient for the tissue slice (Kt), deducing Kf and Kt/Kf is enough (Line 12-17, section 2.6 and 2.7). But the paper also covers estimation of Q/K and the oxygen partial pressure curve. Fick’s-law Model of Diffusion is more about Q/K and the oxygen partial pressure curve. So I suggest the authors rethink about the title and the scope of paper.

4. Fick’s-law Model of Diffusion in Ref[1] is the critical foundation of this paper. Ref [1] only has a few citations since the publication in 2012. Are there any other available models? Could the authors provide more background of diffusion models?

5 The cutoff of Kt/Kf (= 0.725) obviously affected the final aggregate (mean ± standard deviation) statistic (Eq 28). The authors should do more justification on their selection of cutoff

Minor:

1. The authors use different terms: rms discrepency (e.g. Line 238), rms difference (e.g. Line 234), rmse(Fig 6). It's better to be consistent.

2. The result section has little discussion on three types of tension profiles (Parabolic/cosh/mixed). Table A1 gives Flat to indicate paracosh or parabolic. What are their distributions in samples? Does Flat affect Kt estimation?

Reviewer 2 Report

This work discuss the Krogh coefficient for thin slices of rodent cortical tissue. Particularly, it utilised the Fick's-law diffusion-consumption model. Some findings are reported at room temperature and some limitations are discussed. Here are some minor comments for consideration:

1. The reference is weak. I would expect a more recent literature list.

2. Are there any other work that could be the modelling method besides Ivanova and Simeonov 2012?

3. It would expect a consolidated conclusion except the large scatter about the mean value for Kt determinations. Currently the results appear to be weak.

4. What would be the significance when the experiments are conducted a realistic scenario, such as truly stationary fluid boundary layer? Please elaborate the difference for better clarification.

Reviewer 4 Report

The manuscript lacks proper order and presentation. It cannot be accepted in its current format. The manuscript needs to be rewritten with proper presentation to make it comprehensible for the readers. There also needs to be appropriate connections between the different sections. Here are some comments that can be used to better present the work for future submission.

  • The manuscript thoroughly lacks references. Citations are needed in many sections of the manuscript. For example - in line number 43-44 the authors wrote “A wide range of values of Krogh coefficient for biological tissue appear in the literature” followed by no citations.

  • The entire introduction just has one reference. 

  • The materials and methods section in the manuscript describes the composition of artificial cerebrospinal fluid. No references have been provided for this composition whatsoever. 

  • The introduction fails to explain the background of the problem and the work presented in the manuscript.

  • A graphical abstract will be helpful for the reader to understand the idea of the manuscript.

  • The conclusion can be elaborated and can be represented in a different section.

Apart from the presentation and structure here are some other ambiguity in the manuscript:

  • The first paragraph of the introduction addresses a problem with the flux of essential nutrients being compromised due to the mechanical damage that appears in the tissue due to sectioning. Has this issue been addressed anywhere in the manuscript?

  • As far as blood flow to the brain is concerned, rather than lacking a blood flow (line 31 in manuscript), the blood brain barrier is well developed and is highly modulated to allow proper supply of nutrients to the brain cells and tissue. The story here remains ambiguous and unconvincing.

Round 2

Reviewer 1 Report

The authors have addressed my major and minor concerns. I have no more concerns. Only one place needs the authors' double check:

In Line 692, the variance in Kt should be 16% as said in Line 633. The 26% variance should be for the Kt/Kf ratio.